# Malting—A method for modifying volatile composition of black, brown and green lentil seeds

**Alan Gasiński** *, Joanna Kawa-Rygielska

Faculty of Biotechnology and Food Science, Department of Fermentation and Cereals Technology, Wrocław University of Environmental and Life Science, Wrocław, Poland

* alan.gasinski@upwr.edu.pl

## Abstract

Technique of malting legume seeds is not currently widespread among scientists as well as industrial maltsters. However, this method of seed modification is successfully used by humankind for millennia to improve technological parameters, as well as change taste and aroma of various food products. Three lentil cultivars (black, brown and green) were malted (steeped, germinated for three various time periods and then kilned) to produce nine lentil malts. Malting had significant influence on the volatile composition of lentil seeds. Total concentration of volatiles in the green lentils increased and decreased in the case of black and brown lentils after malting procedure. However, most importantly, in every lentil cultivar the contribution of various groups of compounds (such as aldehydes, alcohols, terpenes or ketones) to the overall volatilome was changed due to the malting procedure.

## 1. Introduction

Legumes are one of the oldest cultivated crops and play an important part in the plant-based diet. Legumes are still currently one of the most important sources of protein and energy in the diet of people, as well as farm animals around the world [1,2]. Lentils are a high-protein legumes, which are drought resistant and can be grown in various climates, as they are able to thrive in the warm and cool environments [3]. Furthermore, lentils are present in the most of the cuisines throughout the world and can be used to prepare various dishes, therefore improvement of lentil properties could benefit large quantities of people [4]. Furthermore, improvement of the lentil's nutritional value, organoleptic characteristics and technological properties might have an influence on the origin of many novel, plant-based food products. It would be of great importance, because even raw, unmodified lentils possess vast arrays of bio-active phytochemicals, such as phenolics, phytosterols, phytic acid, saponins, tocopherols and carotenoids, which are beneficial in the prevention of many non-communicable diseases [5–7]. Malting is a process which is used primarily to modify grains and the main reason of malting is increasing its enzymatic activity due to the generation and activation of various enzymes [8]. However, during the malting procedure (which consists of the seed steeping, germination and then kilning) many different processes occur, one of which is substantial change of the

**Data Availability Statement:** Below is the URL and DOI of the whole set of data about volatile compounds analysed in the study, uploaded to the public repository: https://figshare.com/articles/

dataset/Whole_data_volatiles_lentil_malts_xlsx/ 24013944 https://doi.org/10.6084/m9.figshare. 24013944.v1.

**Funding:** A.G received award "Innowacyjny Doktorat", no. V, project number N070/0009/21 from Wrocław University of Environmental and Life Sciences (Poland). Funds received were used to buy standards and analytes. APC of the article is co-financed by Wrocław University of Environmental and Life Sciences. Funders had no role in study design, data collection and analysis, decision to publish or preparation of the manuscript.

**Competing interests:** The authors have declared that no competing interests exist.

volatile composition of the malted seed material. Changes in the volatile composition occur due to the physiological changes in the germinating grain and to thethermal process of drying at the end of the malting procedure. Most malted grains acquire aroma which is described as cookie-like, bread-like, toffee-like, nutty and even caramel, chocolate-like, roasted, or coffee-like, depending on the time and temperatures of the kilning process [9]. As the aroma of legumes (raw, boiled or canned) is often not recognized as 'pleasant' or 'tasty' by most consumers, it seems that the procedures which can improve aroma of the legumes (such as malting, typically used for the modification of the barley grains) might be of interest for the farmers and food producers alike [10]. In the previous studies about malting legumes, tin was determined, that soybean, lentil, vetch, peas and chickpeas can be malted in the conditions typical for malting barley, but produced malts were characterised with unadequate technological properties. Mashes produced from these malts had not saccharified, filtered very slowly and yielded low volume of the wort [11]. However, consequent study about malting lentils and beans have shown that, despite poor brewhouse efficiency, malting procedure carefully tailored for the legume seeds might improve their friability as well as reduce concentration of various anti-nutritonal components, such as phytic acid or raffinose-family oligosaccharides [12,13]. In this study, a malting process was applied to three different varieties of *Lens culinaris*, to determine, whether such a simple procedure, which might be implemented into food production process virtually anywhere in the world might significantly change volatile compositions of lentil seeds.

## 2. Materials and methods

### 2.1. Materials

**2.1.1. Raw material.**   Plant material used in this study were seeds of three lentil (*Lens culinaris*) varieties, each with different colour: black lentil of Beluga variety (BL) with green cotyledon colour, brown lentil of variety Firat 87 (BR) with orange cotyledon colour and green lentil of variety Eston (GR) with green cotyledon colour. Lentil seeds were acquired from BioPlanet company (Leszno near Warsaw, Poland). Lentils after harvest, prior to the study were stored in the silo for nine months. Lentils seeds, prior to the malting procedure and analyses, were manually sifted to discard damaged seeds and seeds with visible discolouration. Moisture content of the seeds, before and after malting process, was analysed with the use of MT Moisture Analyser (Brabender, Duisburg, Germany).

**2.1.2 Reagents and standards.**   Reagents used in this study were 2-undecanone (99%, suitable for GC analyses) purchased from the Sigma-Aldrich company (Saint Louis, MO, USA), cyclohexane (99%), sodium chloride (99.8%) and sodium hypochlorite (15%) (Chempur, Piekary Śląskie, Poland). 2-undecanone was mixed with the cyclohexane to produce internal standard with the concentration of one mg of 2-undecanone per one $dm^3$ of cyclohexane. Standards used for identification of volatiles were: 3-octen-2-one ($\geq$98% purity, Supelco); trans-β-ionone ($\geq$97%, Sigma Aldrich); 1-octen-3-ol ($\geq$98%, Sigma Aldrich); 2-ethyl-1-hexanol ($\geq$99%, Sigma Aldrich); 1-octanol ($\geq$99%, Sigma Aldrich); 3,5-dimethylcyclohexanol ($\geq$97.0%, Sigma Aldrich); 1-nonanol ($\geq$98%, Sigma Aldrich); 1,7-octanediol, 3,7-dimethyl ($\geq$98.0%, Lluch Essence, Barcelona, Spain); 1-dodecanol ($\geq$98%, Sigma Aldrich); 3-carene ($\geq$95%, Supelco), (R)-(+)-limonene ($\geq$97%, Sigma Aldrich), eucalyptol ($\geq$99%, Sigma Aldrich); D-carvone ($\geq$96%, Sigma Aldrich); undecane ($\geq$99%, Sigma Aldrich); dodecane ($\geq$99%, Sigma Aldrich); tridecane ($\geq$99%, Sigma Aldrich); tetradecane ($\geq$99%, Sigma Aldrich); pentadecane ($\geq$99%, Sigma Aldrich); hexadecane ($\geq$99%, Sigma Aldrich); octadecane ($\geq$99%, Sigma Aldrich); 4,6-dimethyldodecane ($\geq$98%, SimSon Pharma Limited, Maharashtra, India), 2-pentylfuran ($\geq$98%, Sigma Aldrich); benzothiazole ($\geq$96%, Sigma Aldrich);

2-ethyl-3,5-dimethyl-2-pyrazine (≥95%, Sigma Aldrich); propanoic acid, 2-methyl-, 3-hydroxy-2,2,4-trimethylpentyl ester (≥95%, Toronto Research Chemicals; Toronto, Canada); dodecanoic acid, 1-methylethyl ester (≥98%, Sigma Aldrich).

## 2.2. Methods

### 2.2.1. Malting procedure.

Simplified diagram of the malting procedure is shown in the Fig 1.

Eighty gram portions of black, brown and green lentil were weighed and transferred to perforated, stainless steel malting containers (24 containers for each of the lentil varieties), which were previously disinfected by drying in the UF110 Plus dryer (Memmert GmbH + Co, Schwabach, Germany) for two h at 200˚C and then cooled to room temperature. Containers filled with known mass of lentil were then weighed (the container filled with lentil seeds will be from now on mentioned as the 'malting kit'). Changes in the moisture content of the seeds during the first step of the malting process (steeping) were calculated based on the changing weight of the malting kit, assuming, that the increase in weight of the kit is equal to the quantity of water adsorbed by the seeds. Steeping was executed in the water-air steeping cycle. At the start of the

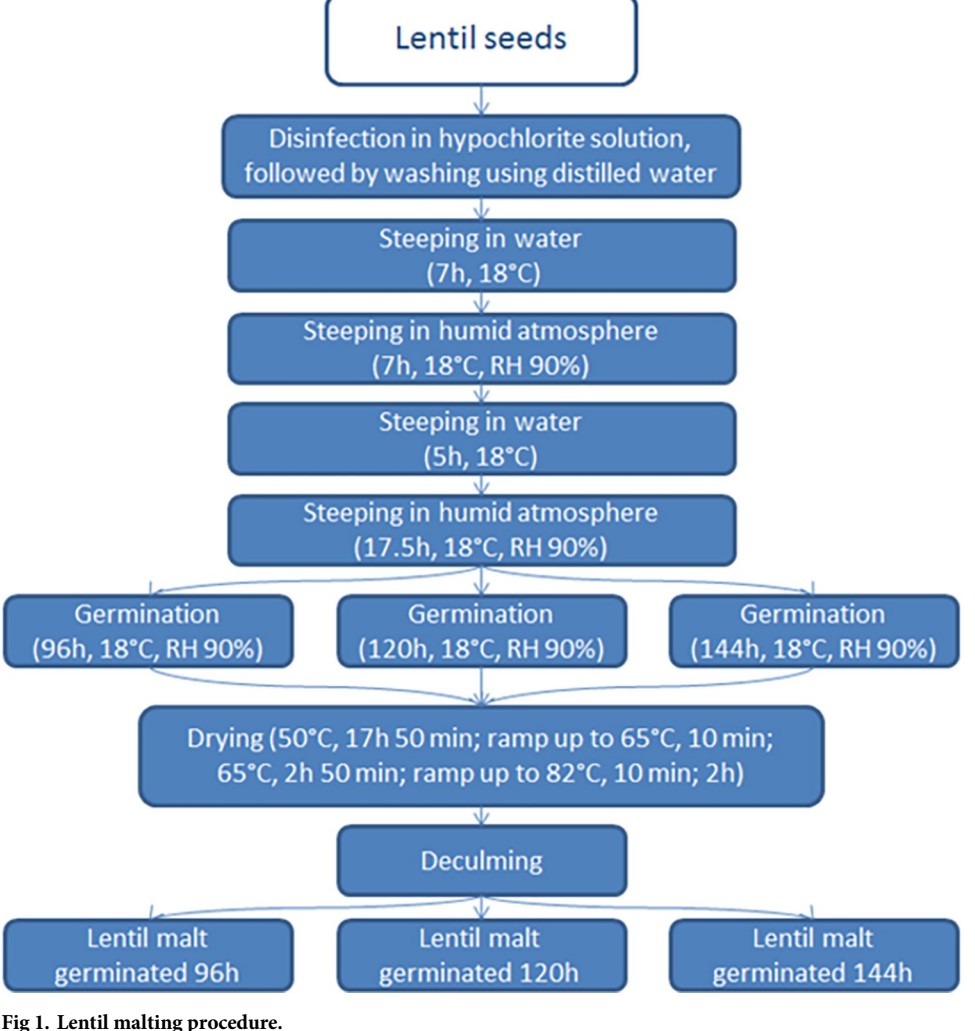

**Fig 1. Lentil malting procedure.**

process, malting kits were submerged in the 1.5% sodium hypochlorite solution for 10 min to surface sterilise the seeds. Malting kits were then removed from the sodium hypochlorite solution and immediately washed three times with distilled water. After this process, malting kits were submerged in tap water (disinfected previously by boiling and cooled) at temperature of 18˚C for seven h, then transferred to the KK 240 Smart Pro germination chamber (with humidity set at 90% relative humidity and temperature set at 18˚C) for 18.5 h; then submerged another time in fresh, disinfected tap water at a temperature of 18˚C for five h; then transferred to the germination chamber (temp. 18˚C, relative humidity 90%) for 17.5 h. After each step, malting kits were weighed to determine changing moisture content of the lentil seeds. At the end of the steeping process, the moisture content of the lentil seeds was equal to 53–55%. Lentils were germinated in the germination chamber with the temperature set at 18˚C and relative humidity 90%. Eight of each of the lentil varieties samples (total of 24 malting kits) were germinated for four days; eight of each of the lentil varieties samples (total of 24 malting kits) were germinated for five days; eight of each of the lentil varieties samples (total of 24 malting kits) were germinated for six days. The germination time used in this study was selected using standard germination time used for production of grain malts [8,14]. After the germination process, each batch of malting kits was dried in the UF110 Plus dryer at the following conditions: 50˚C (17 h and 50 min), ramp up to 65˚C (10 min), 65˚C (two h and 50 min), ramp up to 82˚C (10 min), 82˚C (two h). The malting procedure resulted in production of 9 different lentil malt samples:

- black lentil malt germinated 96 h (four days) (BL4)

- black lentil malt germinated 120 h (five days) (BL5)

- black lentil malt germinated 144 h (six days) (BL6)

- brown lentil malt germinated 96 h (four days) (BR4)

- brown lentil malt germinated 120 h (five days) (BR5)

- brown lentil malt germinated 144 h (six days) (BR6)

- green lentil malt germinated 96 h (four days) (GR4)

- green lentil malt germinated 120 h (five days) (GR5)

- black lentil malt germinated 144 h (six days) (GR6)

The malts, after the drying process, malts of one type from different malting kits were mixed together and transferred to tightly closed containers, to prevent moisture absorption during cooling period. Malts, as well as unmalted lentils, were ground with the use of Bühler Miag disc mill DLFU (Bühler, Uzwil, Switzerland), according to the Analytica EBC 4.5.1 method for the subsequent analysis [15].

**2.2.2. Adsorption of volatile compounds to the solid-phase microextraction fiber.** To perform chromatographic analysis of volatiles present in the lentil seeds and malts, the volatiles had to be adsorbed on the solid phase microextraction fiber (SPME) [16]. Ground malt or seed sample (2.5 g) was transferred to the 20 $cm^3$ headspace vial, followed by the addition of 0.5 g sodium chloride and four $cm^3$ of distilled water. Twenty ng of internal standard (2-undecanone, in the form of 20 $mm^3$ of 2-undecanone in hexane solution) was added to the vial, which was then closed with a magnetic screw-top cap with a septum. SPME holder needle, equipped DVB/CAR/PDMS fiber (50/30μm) (Supelco, Bellefonte, PA, USA) was used to pierce the septum. Vial was positioned on the heatplate set at 80˚C. After 5 min of temperature equilibration, the fiber was extended from the holder needle, to allow adsorption of the volatiles on the fiber surface for 45 min. After adsorption of the volatiles, fiber was retracted into the holder.

**2.2.3. Gas chromatography and mass spectrometry.** Gas chromatography and mass spectrometry of the volatiles was performed using GC-2010 Plus chromatograph coupled with GCMS-QP2010 SE mass spectrometer (Shimadzu, Kyoto, Japan) equipped with ZB-5 column (Phenomenex, Torrance, CA, USA) (30 m length x 0.25 mm internal diameter x 0.25 μm film thickness). Injection port temperature was held at 195˚C. Analyses were carried out with the use of helium as a carrier gas with a flow rate of 1.78 cm$^3$/min and a starting pressure set at 100 kPa. Following program was used for the oven temperature: 40˚C at the beginning; hold for one min, ramp up at a rate of eight˚C/min to 195˚C; hold for five min. Ion source temperature was maintained at 250˚C, while interface temperature was at 195˚C. Scanning was carried out in the 35–350 m/z range using 70 mV electron ionisation with the event time equal to 0.3 s (scan speed equal to 1111). Adsorption of volatiles and gas chromatography was performed in triplicate for each of the lentil samples.

## 2.3. Data analysis

Volatile compounds separated from the lentil seeds and lentil malts were identified by mass spectral analysis based on NIST17 chemical standard libraries and spectra of authentic chemicals, whenever possible, comparison of retention time with retention time of authentic chemicals and by confirmation by comparison of retention indices with Kovats standards. If the authentic chemical standard was not available, the similarity search based on the NIST libraries had to be at least 95% to determine the compound as identified. If the compound was identified with the use of authentic chemicals, the retention time of the authentic chemical could not deviate by more than 0.05 min from the retention time of the authentic chemical sampled on the same column, with the same temperature program. Kovats indices (KI) were used to confirm the identification: if the KI of the identified peak deviated by more than 10 from average KI for the compound declared in the NIST Webbook, then the compound was classified as unidentified and not recorded. Chromatographic peaks were integrated with the use of Shimadzu PostRun Analysis program (Shimadzu, Kyoto, Japan). The results of the analysis of antioxidative activity and phenolic content of lentil seeds and lentil malts were statistically analysed in the Statistica 12.5 program from Statsoft (Tulsa, OK, USA) using two-way ANOVA with variables: variety of lentil and length of germination (unmalted samples were described as having zero days of germination) with $\alpha = 0.05$ using Tukey test.

## 3. Results and discussion

### 3.1. Concentration of volatile compounds in the lentil seeds and lentil malts

Gas chromatography and mass spectrometry allowed for identification and quantification of 50 volatile compounds. The largest group of compounds were aldehydes (18 compounds) and hydrocarbons (nine compounds). Smaller groups of compounds were alcohols (seven compounds), terpenes (sex compounds), ketones (four compounds) and other minor constituents, such as esters, furans, pyrazines and sulphur compounds (total of six compounds). Concentration of total identified volatiles was in the range of 33.10–114.58 ppb, with the lowest for BL and the highest for GR. Malting resulted in a decrease of total concentration of volatiles for green lentil, with minor decrease (by 4.6%) for GR4 and major reduction of volatiles for GR5 and GR6 (by 34.5% and 58.5%, consecutively). However, malting procedure increased the total concentration of volatiles in the samples produced from black and brown lentil. The highest increase of the volatile compounds content for the black lentil was noted between sample BL and BL4, where it increased from 33.10 ppb to 44.11 ppb (increase of 32.8%). Extension of the

germination time resulted in decrease of the total volatile content in the malts from black lentil (37.53 ppb for BL5 and 35.64 ppb for BL6). Different result can be seen in the malts produced from the brown lentil. Sample BR4 was characterised with the lowest concentration of volatiles (35.74 ppb), which increased with the malting, albeit the highest concentration of volatiles was determined in the sample BR5 (50.71 ppb). The main difference between various impact of the malting on the total concentration of volatiles in malts produced from the lentils of different colour seems to be vastly different content of volatiles in the particular type of lentil seeds. The main compounds in the GR, which are absent or in far lower concentration than in BR or BL are eucalyptol, limonene, 3-carene and 1-octen-3-ol. Changes of these constituents are more broadly discussed in the sections 3.1.2 and 3.1.4. Additionally, malting changes the contribution of various groups of volatiles (such as aldehydes, alcohols, esters, ketones, terpenes, furans and pyrazines) in the volatilome of the lentil malts and these results are shown on the Figs 2–4.

**3.1.1. Concentration of aldehydes in the lentil seeds and lentil malts.** Most of the volatiles present in the lentil and lentil malt samples were aldehydes (Table 1). Aldehydes were most abundant group of volatiles in eight out of nine malted lentil samples (with the exception of GR5), where they constituted from 32.24% to 58.75% of all identified volatiles. Malting resulted in the increase of concentration of aldehydes in most of the samples, from 28% to 96%, compared to the unmalted lentil, with the exception of GR6, where the total amount of aldehydes was reduced only by small margin of 0.3 ppb (1.3%). All of the aldehydes present in the lentils and lentil malts have been previously identified in the lentil seeds or lentil flours by various researchers in the previous years [17–19]. Many differences could be seen between various lentil cultivars analysed in this study. Benzaldehyde, compound characterised with almond-like aroma, produced by the Strecker degradation of phenylalanine amino acid was present only in the malted brown lentil samples [20]. This result suggests, that proteases in the brown lentil allowed for the release of phenylalanine during the malting procedure. Benzene

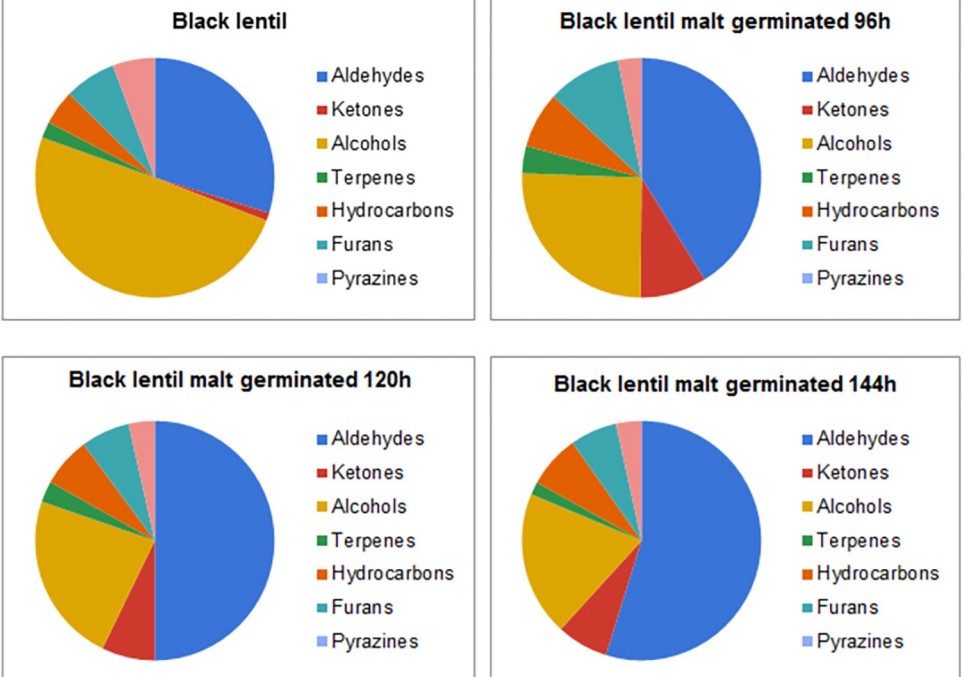

**Fig 2. Contribution of various chemical groups in the volatilome of black lentil and black lentil malts.**

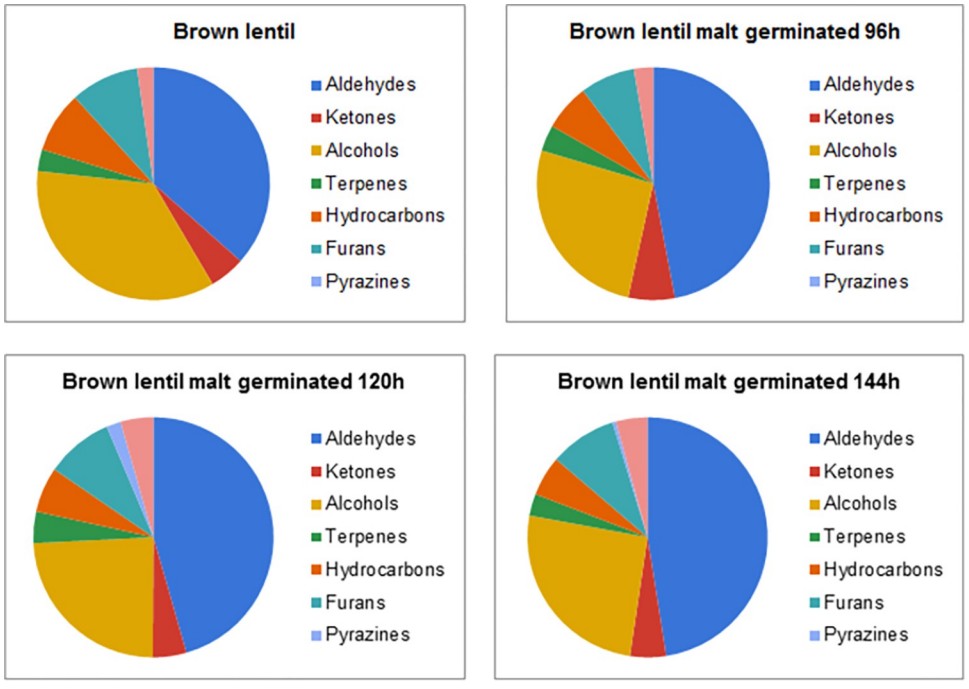

**Fig 3. Contribution of various chemical groups in the volatilome of brown lentil and brown lentil malts.**

acetaldehyde (known also as phenylacetaldehyde), characterised with floral, sweet and honey-like flavour, is a compound, which was not found in the unmalted BR, BL and GR samples, but was present in all the lentil malts. Benzene acetaldehyde was similarly not detected in raw legume products by other scientists such as Wang et al. [21] and Murat et al. [22], but was

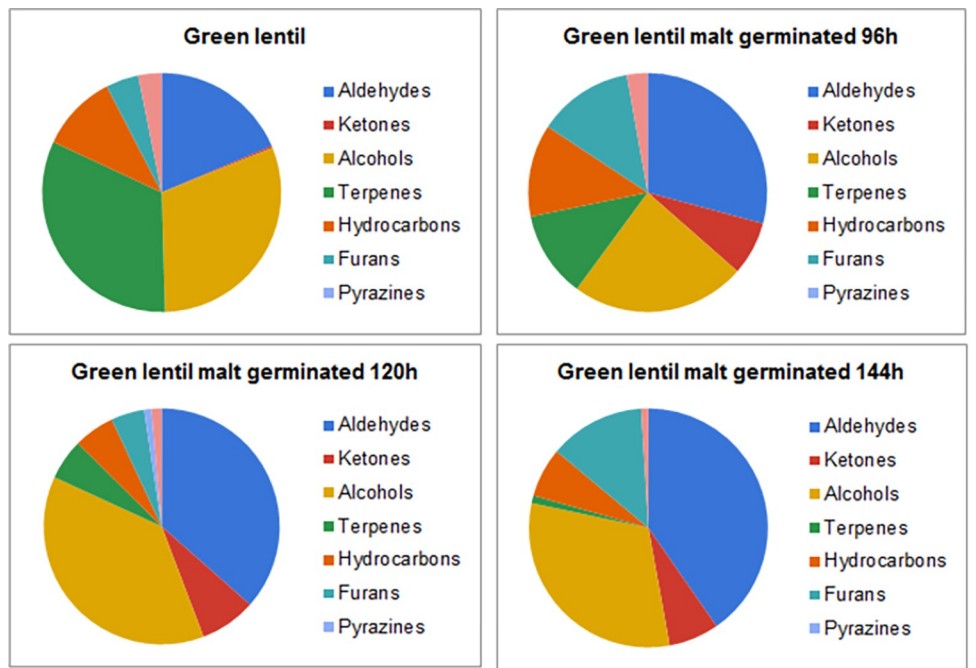

**Fig 4. Contribution of various chemical groups in the volatilome of green lentil and green lentil malts.**

**Table 1. Concentration of aldehydes in lentils and malted lentils.**

| | Compound[1] | BL | BL4 | BL5 | BL6 | BR | BR4 | BR5 | BR6 | GR | GR4 | GR5 | GR6 |
|---|---|---|---|---|---|---|---|---|---|---|---|---|---|
| | | ppb | ppb | ppb | ppb | ppb | ppb | ppb | ppb | ppb | ppb | ppb | ppb |
| 1 | Benzaldehyde | 0.00 b,B | 0.00 b,A | 0.00 b,A | 0.00 b,A | 0.00 a,B | 7.48 ± 0.92 a,A | 7.34 ± 0.88 a,A | 7.96 ± 1.05 a,A | 0.00 b,B | 0.00 b,A | 0.00 b,A | 0.00 b,A |
| 2 | Octanal | 0.00 c,B | 0.00 c,C | 0.00 c,D | 0.00 c,A | 0.00 b,B | 0.42 ± 0.09 b,C | 0.19 ± 0.08 b,D | 0.76 ± 0.21 b,A | 0.87 ± 0.17 a,B | 0.00 a,C | 0.00 a,D | 1.03 ± 0.25 a,A |
| 3 | Benzeneacetaldehyde | 0.00 b,D | 0.69 ± 0.18 b,D | 0.87 ± 0.25 b,A | 1.23 ± 0.21 b,B | 0.00 b,D | 0.89 ± 0.18 b,C | 1.09 ± 0.24 b,A | 0.53 ± 0.13 b,B | 0.00 a,D | 0.92 ± 0.18 a,C | 2.52 ± 0.54 a,A | 0.94 ± 0.31 a,B |
| 4 | 2-Octenal, (E)- | 0.61 ± 0.18 b,C | 0.72 ± 0.22 b,B | 0.90 ± 0.24 b,B | 1.17 ± 0.29 b,A | 0.91 ± 0.28 a,C | 0.44 ± 0.16 a,B | 1.09 ± 0.21 a,B | 1.21 ± 0.43 a,A | 0.00 c,C | 1.50 ± 0.18 c,B | 0.59 ± 0.12 c,B | 0.68 ± 0.21 c,A |
| 5 | Nonanal | 2.70 ± 0.41 c,B | 5.15 ± 0.98 c,A | 4.40 ± 1.03 c,C | 4.21 ± 0.77 c,D | 5.48 ± 1.12 b,A | 4.81 ± 0.68 b,B | 4.84 ± 0.83 b,C | 4.44 ± 0.91 b,D | 11.75 ± 2.18 a,B | 15.28 ± 2.84 a,A | 7.61 ± 1.96 a,C | 5.91 ± 1.04 a,D |
| 6 | 2-Nonenal, (Z)- | 0.00 b,C | 0.14 ± 0.07 b,C | 0.00 b,C | 0.00 b,B | 0.00 c,C | 0.00 c,A | 0.00 c,C | 0.00 c,B | 0.00 a,C | 1.25 ± 0.38 a,A | 0.00 a,C | 0.19 ± 0.08 a,B |
| 7 | Trans-2-nonenal | 3.22 ± 0.72 b,D | 8.94 ± 1.15 b,A | 8.27 ± 0.98 b,B | 7.58 ± 1.15 b,C | 3.17 ± 0.61 c,D | 6.41 ± 1.12 c,A | 6.23 ± 1.24 c,B | 6.24 ± 0.93 c,C | 1.72 ± 0.41 a,D | 11.57 ± 2.81 a,A | 10.41 ± 1.54 a,B | 8.39 ± 0.99 a,C |
| 8 | Decanal | 1.81 ± 0.44 b,A | 1.75 ± 0.31 b,C | 2.26 ± 0.52 b,C | 2.50 ± 0.72 b,B | 2.65 ± 0.81 b,A | 1.48 ± 0.46 b | 1.51 ± 0.37 b,C | 1.98 ± 0.48 b,B | 4.41 ± 0.86 a,A | 2.55 ± 0.61 a,D | 2.53 ± 0.84 a,C | 2.20 ± 0.68 a,B |
| 9 | 2,4-Nonadienal, (E,E)- | 0.00 b,C | 0.15 ± 0.07 b,C | 0.00 b,B | 0.00 b,C | 0.00 a,C | 0.11 ± 0.06 a,B | 0.16 ± 0.09 a,A | 0.00 a,C | 0.00 a,C | 0.00 a,C | 0.26 ± 0.12 a,A | 0.00 a,C |
| 10 | 2-Decenal, (E)- | 0.35 ± 0.11 a,C | 0.61 ± 0.15 a,B | 0.54 ± 0.18 a,A | 0.49 ± 0.16 a,A | 0.39 ± 0.12 a,C | 0.39 ± 0.14 a,B | 0.56 ± 0.21 a,A | 0.50 ± 0.19 a,A | 0.38 ± 0.12 b,C | 0.48 ± 0.21 b,B | 0.53 ± 0.18 b,A | 0.55 ± 0.21 b,A |
| 11 | Benzeneacetaldehyde,.alpha.-ethylidene- | 0.00 b,D | 0.19 ± 0.08 b,A | 0.00 b,B | 0.13 ± 0.06 b,C | 0.00 b,D | 0.12 ± 0.04 b,A | 0.14 ± 0.05 b,B | 0.00 b,C | 0.00 a,D | 0.53 ± 0.18 a,A | 0.63 ± 0.21 a,B | 0.42 ± 0.18 a,C |
| 12 | Undecanal | 0.63 ± 0.21 a,C | 0.37 ± 0.11 a,D | 0.41 ± 0.15 a,B | 0.50 ± 0.18 a,A | 0.12 ± 0.03 b,C | 0.00 b,D | 0.39 ± 0.12 b,B | 0.42 ± 0.14 b,A | 0.00 c,C | 0.00 c,D | 0.17 ± 0.05 c,B | 0.17 ± 0.09 c,A |
| 13 | 2-Octenal, 2-butyl- | 0.00 b,C | 0.00 b,C | 0.00 b,C | 0.00 b,C | 0.00 b,C | 0.00 b,D | 0.00 b,C | 0.00 b,C | 0.00 a,C | 0.00 a,C | 0.94 ± 0.31 a,A | 0.54 ± 0.17 a,B |
| 14 | Dodecanal | 0.00 a,B | 0.00 a,A | 0.21 ± 0.09 a,A | 0.62 ± 0.21 a,A | 0.00 b,B | 0.00 b,B | 0.38 ± 0.11 b,A | 0.00 b,A | 0.00 c,B | 0.00 c,B | 0.00 c,A | 0.00 c,A |
| 15 | 5-Methyl-2-phenyl-2-hexenal | 0.00 b,A | 0.00 b,A | 0.00 b,C | 0.00 b,B | 0.00 b,A | 0.00 b,A | 0.00 b,C | 0.00 b,B | 0.00 a,B | 0.33 ± 0.14 a,A | 0.00 a,C | 0.08 ± 0.05 a,B |
| 16 | Tridecanal | 0.45 ± 0.19 a,A | 0.50 ± 0.16 a,A | 0.74 ± 0.28 a,A | 0.86 ± 0.29 a,B | 0.64 ± 0.18 b,A | 0.45 ± 0.18 b,C | 0.50 ± 0.21 b,A | 0.52 ± 0.25 b,B | 0.89 ± 0.32 a,A | 0.60 ± 0.16 a,C | 0.82 ± 0.25 a,A | 0.29 ± 0.08 a,B |
| 17 | Tetradecanal | 0.22 ± 0.12 a,D | 0.24 ± 0.10 a,C | 0.51 ± 0.22 a,A | 0.52 ± 0.19 a,B | 0.26 ± 0.09 b,D | 0.24 ± 0.09 b,C | 0.29 ± 0.11 b,A | 0.18 ± 0.08 b,B | 0.00 c,D | 0.17 ± 0.08 c,C | 0.40 ± 0.21 c,A | 0.16 ± 0.08 c,B |
| 18 | Pentadecanal | 0.67 ± 0.28 b,A | 0.76 ± 0.27 b,C | 1.02 ± 0.34 b,B | 1.13 ± 0.39 b,D | 0.89 ± 0.32 c,A | 0.85 ± 0.25 c,C | 0.67 ± 0.21 c,B | 0.60 ± 0.19 c,D | 2.36 ± 0.81 a,A | 0.97 ± 0.33 a,C | 1.24 ± 0.38 a,B | 0.52 ± 0.16 a,D |
| | **Total aldehydes** | 10.67 | 20.21 | 20.12 | 20.94 | 14.52 | 24.09 | 25.38 | 25.33 | 22.38 | 36.14 | 28.67 | 22.08 |
| | **% of all volatiles** | 32.24% | 45.81% | 53.62% | 58.75% | 40.62% | 50.96% | 50.04% | 52.76% | 19.53% | 33.10% | 38.28% | 46.54% |

[1] Abbreviations are as follows: GL4—malt from green lentil germinated 4 days, GL5—malt from green lentil germinated 5 days, GL6—malt from green lentil germinated 6 days, BRL4—malt from brown lentil germinated 4 days, BRL5—malt from brown lentil germinated 5 days, BRL6—malt from brown lentil germinated 6 days, BLL4—malt from black lentil germinated 4 days, BLL5—malt from black lentil germinated 5 days, BLL6—malt from black lentil germinated 6 days. Values are expressed as means (n = 3) ± standard deviation. Various small letters (a, b, c) indicate homogenous groups according to the variable 'variety', various capital (A, B, C, D) letters indicate homogenous groups according to the variable 'days of germination'(Tukey test, α = 0.05).

identified in the processed legume products, such as flours and protein isolates. Malting increased the concentration of trans-2-nonenal, which is produced by the oxidation of free fatty acids, in all the lentil varieties. In the malting industry, this compound is viewed negatively, as the aroma of trans-2-nonenal is characterised with the stale beer (possessing so-called 'cardboard' flavour) [23], albeit study performed by the Shi et al. [24] about beverage produced from the legume seeds (soymilk) showed, that experts in sensory analysis preferred legume products with higher concentration of trans-2-nonenal. This result might suggest that legume malts are not the best substrate for the production of wort and beer, albeit might find their uses in the production of the milk substitutes.

**3.1.2. Concentration of alcohols in the lentil seeds and lentil malts.** Alcohols were the second largest group of volatiles in the lentil malts, in which the volatile composition consisted from 20.96% (BL6) to 39.45% (GR5) of alcohols (Table 2). Alcohols were main volatile components in the sample BL (53.77%), as well as constituted second largest groups of compounds in the BR (39.15%) and GR (32.19%) sample. Malting had almost no impact on the total concentration of alcohols in the malts prepared from the brown lentil (13.99 ppb in BR, 13.40–13.64 ppb in the malted samples). The concentration of alcohols in black and green lentil, however, was decreased by malting. Total reduction of alcohols in the black lentil was lowest for BL4, in which the decrease by 5.29 ppb was noted (reduction by 29.7% of total alcohols) and the highest for BL6, in which total concentration of alcohols was reduced by 10.33 ppb (reduction by 58%). Similar results can be seen in the green lentil malts: the lowest decrease, by 7.61 ppb (20.6%), for GR4, and the highest, by 19.8 ppb (53.7%), for GR6. However, close attention ought to be put to the concentration of two important alcohols, which were identified in all the analysed lentil and lentil malt samples. 1-octen-3-ol is a compound with characteristic, unpleasant mushroom aroma, which is present in various legumes, such as soybeans and legume-based food products [25,26]. Malted black and green lentils were characterised with lower concentration of this compound, especially samples BL6 and GR6, in which concentration of 1-octen-3-ol was reduced by more than a 50%. Second alcohol, characterised with earthy, pea-like note, which was reduced in the course of malting, was 1-nonanol. Malting allowed for the reduction of this compound in unmalted lentils by 2.04 ppb (79.4%) in case of GL6; by 0.76 ppb (65.5%) in the of BL6 and by 0.29 ppb (36.7%) in case of BR6. These results suggest that lentil malts can be used to produce lentil-based products for the groups of population which try to avoid legume-based products, because of their aroma.

**3.1.3. Concentration of hydrocarbons in the lentil seeds and lentil malts.** Volatilome of the lentils seeds and lentil malts contained vast array of hydrocarbons, albeit most of the compounds from this chemical group were present only in small amounts (below one ppb) (Table 3). Sample BL was characterised with lowest concentration of hydrocarbons (1.68 ppb) and malting seemed to increase total concentration of hydrocarbons, as well as their share in the volatilome of black malts, with the highest increase (by 121.4% of terpenes) noted for BL4. In contrast, malting did not change significantly concentration of hydrocarbons in BR4 and BR5 and decreased total concentration of hydrocarbons in the sample BR6 by 0.5 ppb (by 14.7% of all terpenes in the sample). The greatest change of the concentration of hydrocarbons in the course of malting could be seen in the malts prepared from the green lentil. Sample GL4 possessed 2.92 ppb of the hydrocarbons more than GL (increase of 23.2%), but increasing germination time to five or six days resulted in far lower concentration of hydrocarbons in GL5 (lower by 8.17 ppb, decrease by 65%) and GL6 (lower by 9.91 ppb, decrease by 70.8%) than in GL. In the previous years, scientists such as Rajhi et al. [17] and Bi et al. [27] have detected small amounts of hydrocarbons in the volatilome of lentils and various, different legume seeds. It was speculated, on the basis of the research conducted by the Oomah et al. [28] about volatilome of beans, that most hydrocarbons (specifically n-alkanes) in the volatile composition of

**Table 2. Concentration of alcohols in lentils and malted lentils.**

| | | BL | BL4 | BL5 | BL6 | BR | BR4 | BR5 | BR6 | GR | GR4 | GR5 | GR6 |
|---|---|---|---|---|---|---|---|---|---|---|---|---|---|
| | | ppb | ppb | ppb | ppb | ppb | ppb | ppb | ppb | ppb | ppb | ppb | ppb |
| 1 | 1-Octen-3-ol | 14.22 ± 2.31 c, A | 9.96 ± 2.08 c, B | 8.20 ± 1.14 c, C | 6.25 ± 0.98 c, D | 11.28 ± 2.98 b, A | 12.17 ± 2.08 b, B | 11.26 ± 1.58 b, C | 10.87 ± 1.85 b, D | 28.73 ± 3.11 a, A | 26.26 ± 2.51 a, B | 24.89 ± 2.08 a, C | 14.20 ± 1.98 a, A |
| 2 | 1-Hexanol, 2-ethyl- | 0.00 b, A | 0.00 b, B | 0.00 b, B | 0.00 b, B | 0.00 b, A | 0.00 b, B | 0.00 b, B | 0.00 b, B | 1.09 ± 0.41 a, A | 0.00 a, B | 0.00 a, B | 0.00 a, B |
| 3 | 1-Octanol | 1.57 ± 0.39 c, A | 0.00 c, D | 0.00 c, C | 0.30 ± 0.12 c, B | 1.24 ± 0.42 b, A | 0.00 b, D | 0.64 ± 0.28 b, C | 1.33 ± 0.51 b, B | 3.14 ± 0.92 a, A | 0.00 a, D | 1.82 ± 0.43 a, C | 1.43 ± 0.28 a, B |
| 4 | Cyclohexanol, 3,5-dimethyl- | 0.00 c, D | 1.27 ± 0.32 c, A | 0.00 c, B | 0.00 c, C | 0.41 ± 0.11 b, D | 0.46 ± 0.18 b, A | 0.67 ± 0.21 b, B | 0.48 ± 0.15 b, C | 0.00 a, D | 0.72 ± 0.30 a, A | 1.22 ± 0.22 a, B | 0.59 ± 0.18 a, C |
| 5 | 1-Nonanol | 1.16 ± 0.43 b, A | 0.65 ± 0.28 b, B | 0.52 ± 0.21 b, C | 0.40 ± 0.09 b, D | 0.79 ± 0.19 c, A | 0.40 ± 0.17 c, B | 0.43 ± 0.18 c, C | 0.50 ± 0.22 c, D | 2.57 ± 0.94 a, A | 1.47 ± 0.76 a, B | 0.86 ± 0.25 a, C | 0.53 ± 0.21 a, D |
| 6 | 1,7-octanediol, 3,7-dimethyl | 0.42 ± 0.18 b, A | 0.23 ± 0.05 b, B | 0.20 ± 0.08 b, C | 0.18 ± 0.09 b, D | 0.09 ± 0.06 b, A | 0.28 ± 0.17 b, B | 0.30 ± 0.12 b, C | 0.31 ± 0.11 b, D | 0.99 ± 0.23 a, A | 0.82 ± 0.24 a, B | 0.39 ± 0.15 a, C | 0.20 ± 0.08 a, D |
| 7 | 1-Dodecanol | 0.42 ± 0.18 a, A | 0.39 ± 0.15 a, D | 0.38 ± 0.17 a, B | 0.34 ± 0.21 a, C | 0.18 ± 0.06 c, A | 0.09 ± 0.05 c, D | 0.14 ± 0.06 c, B | 0.15 ± 0.06 c, C | 0.36 ± 0.11 b, A | 0.00 b, D | 0.39 ± 0.17 b, B | 0.12 ± 0.04 b, C |
| | **Total alcohols** | 17.80 | 12.51 | 9.29 | 7.47 | 13.99 | 13.40 | 13.44 | 13.64 | 36.88 | 29.27 | 29.55 | 17.08 |
| | **% of all volatiles** | 53.77% | 28.35% | 24.77% | 20.96% | 39.15% | 28.34% | 26.50% | 28.41% | 32.19% | 26.81% | 39.45% | 36.01% |

[1] Abbreviations are as follows: GL4—malt from green lentil germinated 4 days, GL5—malt from green lentil germinated 5 days, GL6—malt from green lentil germinated 6 days, BRL4—malt from brown lentil germinated 4 days, BRL5—malt from brown lentil germinated 5 days, BRL6—malt from brown lentil germinated 6 days, BLL4—malt from black lentil germinated 4 days, BLL5—malt from black lentil germinated 5 days, BLL6—malt from black lentil germinated 6 days. Values are expressed as means (n = 3) ± standard deviation. Various small letters (a, b, c) indicate homogenous groups according to the variable 'variety', various capital (A, B, C) letters indicate homogenous groups according to the variable 'days of germination'(Tukey test, α = 0.05).

**Table 3. Concentration of terpenes and ketones in lentils and malted lentils.**

| | BL (ppb) | BL4 (ppb) | BL5 (ppb) | BL6 (ppb) | BR (ppb) | BR4 (ppb) | BR5 (ppb) | BR6 (ppb) | GR (ppb) | GR4 (ppb) | GR5 (ppb) | GR6 (ppb) |
|---|---|---|---|---|---|---|---|---|---|---|---|---|
| 3-Carene | 0.00 b,A | 0.00 b,B | 0.00 b,B | 0.00 b,B | 0.00 b,B | 0.00 b,B | 0.00 b,B | 0.00 b,B | 3.24 ± 0.84 a,A | 0.00 a,A | 0.00 a,A | 0.00 a,A |
| Limonene | 0.00 c,B | 0.97 ± 0.31 c,A | 0.45 ± 0.12 c,C | 0.00 c,D | 0.00 b,D | 0.71 ± 0.28 b,A | 1.08 ± 0.33 b,C | 0.68 ± 0.21 b,D | 10.49 ± 1.16 a,B | 12.81 ± 2.08 a,A | 3.17 ± 0.69 a,C | 0.00 a,D |
| Eucalyptol | 0.00 b,A | 0.00 b,B | 0.00 b,B | 0.00 b,B | 0.00 b,A | 0.00 b,B | 0.00 b,B | 0.00 b,B | 23.63 ± 2.49 a,A | 0.00 a,B | 0.00 a,B | 0.00 a,B |
| D-Carvone | 0.50 ± 0.11 a,A | 0.13 ± 0.06 a,B | 0.00 a,B | 0.00 a,B | 0.00 b,B | 0.00 b,B | 0.00 b,B | 0.00 b,B | 0.00 b,B | 0.00 b,B | 0.00 b,B | 0.00 b,B |
| Gamma-Muurolene | 0.28 ± 0.15 c,C | 0.44 ± 0.18 c,A | 0.46 ± 0.21 c,A | 0.42 ± 0.18 c,D | 0.76 ± 0.28 b,C | 0.76 ± 0.25 b,A | 0.89 ± 0.31 b,B | 0.69 ± 0.17 b,D | 0.88 ± 0.18 a,C | 1.61 ± 0.34 a,A | 0.75 ± 0.25 a,B | 0.41 ± 0.17 a,D |
| .tau.-Cadinol | 0.00 b,A | 0.26 ± 0.13 b,A | 0.23 ± 0.11 b,B | 0.24 ± 0.14 b,B | 0.42 ± 0.15 a,A | 0.37 ± 0.12 a,B | 0.39 ± 0.14 a,A | 0.20 ± 0.09 a,B | 0.71 ± 0.25 a,A | 0.00 a,B | 0.47 ± 0.22 a,A | 0.14 ± 0.05 a,B |
| **Total terpenes** | **0.78** | **1.80** | **1.14** | **0.66** | **1.19** | **1.84** | **2.36** | **1.56** | **38.95** | **14.43** | **4.39** | **0.56** |
| **% of all volatiles** | **2.37%** | **4.08%** | **3.04%** | **1.84%** | **3.32%** | **3.90%** | **4.66%** | **3.26%** | **33.99%** | **13.21%** | **5.86%** | **1.18%** |
| 3-Octen-2-one | 0.00 c,D | 0.52 ± 0.21 c,B | 0.16 ± 0.05 c,A | 0.00 c,C | 0.00 b,D | 0.55 ± 0.23 b,B | 0.23 ± 0.12 b,A | 0.43 ± 0.18 b,C | 0.00 a,D | 0.84 ± 0.23 a,B | 1.81 ± 0.54 a,A | 0.84 ± 0.33 a,C |
| 3,5-Octadien-2-one, (E,E)- | 0.00 a,A | 1.61 ± 0.46 a,A | 1.11 ± 0.27 a,B | 0.63 ± 0.18 a,C | 1.62 ± 0.42 b,A | 0.00 b,A | 0.00 b,B | 0.00 b,C | 0.00 c,A | 0.00 c,B | 0.00 c,C | 0.00 c,C |
| 2-(1-Hydroxybut-2-enylidene) cyclohexanone | 0.00 c,D | 2.06 ± 0.54 c,A | 1.42 ± 0.43 c,B | 1.68 ± 0.38 c,B | 0.08 ± 0.05 b,C | 2.50 ± 0.92 b,A | 2.08 ± 0.71 b,B | 1.54 ± 0.38 b,C | 0.00 a,D | 7.64 ± 2.05 a,A | 3.83 ± 1.04 a,B | 2.61 ± 1.15 a,C |
| Trans-.beta.-Ionone | 0.12 ± 0.05 a,A | 0.00 a,B | 0.00 a,B | 0.03 ± 0.02 a,B | 0.08 ± 0.05 a,A | 0.00 a,B | 0.00 a,B | 0.00 a,B | 0.00 b,B | 0.00 b,B | 0.00 b,B | 0.00 b,B |
| **Total ketones** | **0.12** | **4.19** | **2.69** | **2.34** | **1.78** | **3.05** | **2.31** | **1.97** | **0.00** | **8.48** | **5.64** | **3.44** |
| **% of all volailes** | **0.36%** | **9.49%** | **7.16%** | **6.56%** | **4.97%** | **6.46%** | **4.55%** | **4.11%** | **0.00%** | **7.77%** | **7.53%** | **7.26%** |

[1] Abbreviations are as follows: GL4—malt from green lentil germinated 4 days, GL5—malt from green lentil germinated 5 days, GL6—malt from green lentil germinated 6 days, BRL4—malt from brown lentil germinated 4 days, BRL5—malt from brown lentil germinated 5 days, BRL6—malt from brown lentil germinated 6 days, BLL4—malt from black lentil germinated 4 days, BLL5—malt from black lentil germinated 5 days, BLL6—malt from black lentil germinated 6 days. Values are expressed as means (n = 3) ± standard deviation. Various small letters (a, b, c) indicate homogenous groups according to the variable 'variety', various capital (A, B, C) letters indicate homogenous groups according to the variable 'days of germination'(Tukey test, α = 0.05).

legumes are a product of lipid oxidation. Different concentrations of hydrocarbons in the malts of different varieties of lentils might indicate that the composition of lipids, as well activity of lipases in these lentils is very different. Further study concentrating on the changes in the lipid fraction of lentils during germination and drying are needed to confirm and clarify these hypotheses.

**3.1.4. Concentration of terpenes in the lentil seeds and lentil malts.** Terpenes were mostly present only in the GL and green lentil malts (Table 4). Terpenes constituted only 2.37% of all volatiles in BL and 3.32% of BR. Malting had not changed significantly the contribution of terpenes to the total volatilome of the malt in the case of black and brown lentil malts. 1.82% to 4.06% of the black lentil malt volatilome consisted of terpenes and, similarly, volatilome of brown lentil malts constituted from 3.22% to 4.64%. In contrast, volatile composition of GL contained 33.99% (38.95 ppb) of terpenes. It is worth pointing out, that concentration of terpenes in sample GL is higher than concentration of all volatiles in the samples BL and BR (33.10 and 35.74 ppb). It is worth noting, that terpenes (monoterpenes and sesquiterpenes alike) are not very abundant in the legume seeds, but limonene seems to be most dominant compound of this chemical family and previous researches have detected it in the flours prepared from green and red lentil [29,30]. The effect of malting on the concentration of terpenes, especially 3-carene, limonene and eucalyptol is striking: sixday malting reduced concentration of terpenes in GL6 by 98.6%. Eucalyptol and 3-carene were absent in GL4, GL5 and GL6. However, 4-day malting increased concentration of limonene by 2.32 ppb (22.1%) and of gamma-muurolene by 0.86 ppb (83%). However, five day and sixday malting period resulted in significant decrease of these two terpenes, despite their increase by fourday malting period. This result might be due to the tendency of limonene and other terpenes to degrade under prolonged oxidative conditions [31].

**3.1.5. Concentration of ketones in the lentil seeds and lentil malts.** Malting increased total concentration of ketones in all the analysed lentil varieties (Table 4). Ketones were absent or almost absent in unmalted samples BL and GR (0.12 ppb and 0.34 ppb, consecutively). The most significant increase could be noted for the malted green lentil samples, from zero to 3.44–8.48 ppb (with the highest for GR4 and lowest for GR6). Similar phenomenon could be seen in the malts produced from the other lentil varieties. Malts germinated for fourdays malts were characterised with the highest concentration of ketones, however, difference between malts germinated for four, five or six days from black or brown malts was not as considerable, as in green lentil malts. One of the most interesting aspects of malting on the concentration of ketones in the lentils malts is the presence of 2-(1-hydroxybut-2-enylidene)cyclohexanone. This compound is not widely described in the scientific literature; only mention could be found in the work of Moldoveanu [32], as one of the products of thermal degradation of RuBisCO (ribulose-1,5-bisphosphate carboxylase-oxygenase); a plant enzyme which is involved in the carbon fixation process [33]. As the enzymatic activity and metabolic pathways connected to the photosynthesis increase during the steeping and germination of the seeds, this reason of the increased concentration of 2-(1-hydroxybut-2-enylidene)cyclohexanone seems most plausible, but to confirm this, a study concentrating on this particular problem would be needed in the future [34].

**3.1.6. Concentration of minor volatile constituents (furans, pyrazines, sulphur compounds and esters) in the lentil seeds and lentil malts.** Concentration of constituents from four different groups of chemical compounds (furans, pyrazines, sulphur compounds and esters) constituted minority of the total volatilome of lentil seeds and lentil malts (Table 5). Benzothiazole, a sulphur compound, was present only in the malted lentil samples. The main formation pathway of this compound in the food products is related to the non-enzymatic browning reactions between reducing sugars and amino acids occurring in the presence of

**Table 4. Concentration of hydrocarbons in lentils and malted lentils.**

| | | BL (ppb) | BL4 (ppb) | BL5 (ppb) | BL6 (ppb) | BR (ppb) | BR4 (ppb) | BR5 (ppb) | BR6 (ppb) | GR (ppb) | GR4 (ppb) | GR5 (ppb) | GR6 (ppb) |
|---|---|---|---|---|---|---|---|---|---|---|---|---|---|
| 1 | Undecane | 0.00 c, B | 0.00 c, A | 0.00 c, A | 0.00 c, D | 0.00 c, C | 0.50 ± 0.21 b, A | 0.29 ± 0.08 b, D | 0.42 ± 0.18 b, C | 4.06 ± 0.92 a, B | 4.36 ± 1.14 a, A | 0.00 a, D | 0.92 ± 0.34 a, C |
| 2 | Dodecane | 0.00 b, B | 2.04 ± 0.85 b, A | 1.02 ± 0.26 b, C | 0.58 ± 0.13 b, D | 1.18 ± 0.34 b, B | 1.00 ± 0.25 b, A | 0.80 ± 0.28 b, C | 0.57 ± 0.18 b, D | 4.67 ± 0.88 a, B | 4.77 ± 1.05 a, A | 1.92 ± 0.64 a, C | 1.09 ± 0.29 a, D |
| 3 | 4-methyldodecane | 0.21 ± 0.09 c, A | 0.00 c, B | 0.00 c, B | 0.00 c, B | 0.46 ± 0.12 b, A | 0.00 b, B | 0.00 b, B | 0.00 b, B | 1.19 ± 0.32 a, A | 0.00 a, B | 0.00 a, B | 0.00 a, B |
| 4 | Dodecane, 4,6-dimethyl- | 0.17 ± 0.06 c, D | 0.00 c, C | 0.06 ± 0.04 c, C | 0.31 ± 0.15 c, B | 0.32 ± 0.17 b, D | 0.18 ± 0.08 b, A | 0.25 ± 0.11 b, C | 0.26 ± 0.14 b, B | 0.00 a, D | 1.45 ± 0.49 a, A | 0.31 ± 0.13 a, C | 0.32 ± 0.15 a, B |
| 5 | Tridecane | 0.15 ± 0.09 c, C | 0.16 ± 0.08 c, C | 0.11 ± 0.05 c, B | 0.25 ± 0.14 c, B | 0.19 ± 0.08 b, C | 0.27 ± 0.11 b, A | 0.28 ± 0.14 b, B | 0.22 ± 0.15 b, B | 0.45 ± 0.28 a, C | 1.34 ± 0.49 a, A | 0.44 ± 0.18 a, B | 0.38 ± 0.15 a, B |
| 6 | Tetradecane | 0.49 ± 0.11 b, B | 0.90 ± 0.23 b, A | 0.74 ± 0.28 b, A | 0.67 ± 0.18 b, C | 0.56 ± 0.23 c, B | 0.70 ± 0.31 c, A | 0.67 ± 0.28 c, C | 0.52 ± 0.18 c, D | 1.45 ± 0.42 a, B | 1.81 ± 0.49 a, A | 0.96 ± 0.33 a, C | 0.53 ± 0.15 a, D |
| 7 | Pentadecane | 0.32 ± 0.15 c, C | 0.31 ± 0.13 c, C | 0.39 ± 0.16 c, A | 0.46 ± 0.21 c, C | 0.47 ± 0.24 b, C | 0.46 ± 0.27 b, A | 0.55 ± 0.28 b, B | 0.41 ± 0.21 b, C | 0.40 ± 0.16 a, C | 1.39 ± 0.42 a, A | 0.58 ± 0.18 a, B | 0.33 ± 0.12 a, C |
| 8 | Hexadecane | 0.22 ± 0.07 a, B | 0.21 ± 0.11 a, A | 0.27 ± 0.06 a, C | 0.35 ± 0.14 a, D | 0.19 ± 0.09 b, A | 0.22 ± 0.10 b, A | 0.23 ± 0.08 b, C | 0.19 ± 0.05 b, D | 0.35 ± 0.17 a, B | 0.36 ± 0.11 a, A | 0.20 ± 0.09 a, C | 0.09 ± 0.06 a, D |
| 9 | Octadecane | 0.11 ± 0.06 b, C | 0.10 ± 0.04 b, C | 0.11 ± 0.07 b, A | 0.11 ± 0.06 b, B | 0.00 a, C | 0.00 a, C | 0.34 ± 0.15 a, A | 0.28 ± 0.16 a, B | 0.00 c, C | 0.00 c, C | 0.00 c, A | 0.00 c, B |
| | **Total hydrocarbons** | **1.68** | **3.72** | **2.71** | **2.73** | **3.38** | **3.32** | **3.40** | **2.88** | **12.58** | **15.50** | **4.41** | **3.67** |
| | **% of all volatiles** | **5.07%** | **8.43%** | **7.21%** | **7.66%** | **9.45%** | **7.03%** | **6.70%** | **6.00%** | **10.97%** | **14.19%** | **5.89%** | **7.73%** |

[1] Abbreviations are as follows: GL4—malt from green lentil germinated 4 days, GL5—malt from green lentil germinated 5 days, GL6—malt from green lentil germinated 6 days, BRL4—malt from brown lentil germinated 4 days, BRL5—malt from brown lentil germinated 5 days, BRL6—malt from brown lentil germinated 6 days, BLL4—malt from black lentil germinated 4 days, BLL5—malt from black lentil germinated 5 days, BLL6—malt from black lentil germinated 6 days. Values are expressed as means (n = 3) ± standard deviation. Various small letters (a, b, c) indicate homogenous groups according to the variable 'variety', various capital (A, B, C) letters indicate homogenous groups according to the variable 'days of germination'(Tukey test, $\alpha = 0.05$).

**Table 5. Concentration of minor constituents (furans, pyrazines, sulphur compounds and esters) in lentils and malted lentils.**

| | BL | BL4 | BL5 | BL6 | BR | BR4 | BR5 | BR6 | GR | GR4 | GR5 | GR6 |
|---|---|---|---|---|---|---|---|---|---|---|---|---|
| | ppb | ppb | ppb | ppb | ppb | ppb | ppb | ppb | ppb | ppb | ppb | ppb |
| Furan, 2-pentyl- | 0.00 b, B | 0.00 b, A | 0.00 b, B | 0.00 b, B | 0.00 b, B | 0.00 b, A | 0.00 b, B | 0.00 b, B | 0.00 a, B | 1.88 ± 0.54 a, A | 0.00 a, B | 0.00 a, B |
| **Total furans** | 0.00 | 0.00 | 0.00 | 0.00 | 0.00 | 0.00 | 0.00 | 0.00 | 0.00 | 1.88 | 0.00 | 0.00 |
| **% of all volatiles** | 0.00% | 0.00% | 0.00% | 0.00% | 0.00% | 0.00% | 0.00% | 0.00% | 0.00% | 1.72% | 0.00% | 0.00% |
| Pyrazine, 2-ethyl-3,5-dimethyl- | 0.00 c, C | 0.00 c, C | 0.00 c, C | 0.00 c, B | 0.00 a, C | 0.00 a, C | 1.07 ± 0.24 a, A | 0.25 ± 0.12 a, B | 0.00 b, C | 0.00 b, C | 0.79 ± 0.28 b, A | 0.00 b, B |
| **Total pyrazines** | 0.00 | 0.00 | 0.00 | 0.00 | 0.00 | 0.00 | 1.07 | 0.25 | 0.00 | 0.00 | 0.79 | 0.00 |
| **% of all volatiles** | 0.00% | 0.00% | 0.00% | 0.00% | 0.00% | 0.00% | 2.11% | 0.52% | 0.00% | 0.00% | 1.06% | 0.00% |
| Benzothiazole | 0.00 b, D | 0.13 ± 0.05 b, C | 0.15 ± 0.08 b, A | 0.19 ± 0.10 b, B | 0.00 a, D | 0.19 ± 0.09 a, C | 0.29 ± 0.14 a, A | 0.12 ± 0.07 a, B | 0.00 b, B | 0.00 b, C | 0.36 ± 0.14 b, A | 0.07 ± 0.05 b, B |
| **Total sulfur compounds** | 0.00 | 0.13 | 0.15 | 0.19 | 0.00 | 0.19 | 0.29 | 0.12 | 0.00 | 0.00 | 0.36 | 0.07 |
| **% of all volatiles** | 0.00% | 0.29% | 0.41% | 0.54% | 0.00% | 0.40% | 0.57% | 0.24% | 0.00% | 0.00% | 0.48% | 0.15% |
| Propanoic acid, 2-methyl-, 3-hydroxy-2,2,4-trimethylpentyl ester | 0.78 ± 0.25 c, A | 0.68 ± 0.23 c, B | 0.62 ± 0.31 c, C | 0.56 ± 0.28 c, D | 0.80 ± 0.33 a, A | 0.65 ± 0.23 a, B | 1.06 ± 0.34 a, C | 0.78 ± 0.31 a, D | 1.70 ± 0.54 b, A | 1.36 ± 0.28 b, B | 0.00 b, C | 0.00 b, D |
| Pentanoic acid, 2,2,4-trimethyl-3-carboxyisopropyl, isobutyl ester | 1.05 ± 0.28 c, B | 0.73 ± 0.31 c, A | 0.51 ± 0.19 c, C | 0.54 ± 0.19 c, D | 0.09 ± 0.05 b, B | 0.63 ± 0.24 b, A | 1.26 ± 0.34 b, C | 1.12 ± 0.48 b, D | 2.10 ± 0.58 a, B | 2.11 ± 0.63 a, A | 1.09 ± 0.27 a, C | 0.54 ± 0.21 a, D |
| Dodecanoic acid, 1-methylethyl ester | 0.22 ± 0.12 a, D | 0.15 ± 0.08 a, C | 0.30 ± 0.12 a, B | 0.21 ± 0.13 a, A | 0.00 b, D | 0.10 ± 0.07 b, C | 0.14 ± 0.06 b, B | 0.36 ± 0.14 b, A | 0.00 c, C | 0.00 c, C | 0.00 c, B | 0.00 c, A |
| **Total esters** | 2.05 | 1.56 | 1.42 | 1.32 | 0.89 | 1.38 | 2.47 | 2.26 | 3.80 | 3.47 | 1.09 | 0.54 |
| **% of all volatiles** | 6.19% | 3.54% | 3.80% | 3.69% | 2.49% | 2.91% | 4.87% | 4.71% | 3.32% | 3.18% | 1.45% | 1.14% |
| **Total minor constituents** | 2.05 | 1.69 | 1.58 | 1.51 | 0.89 | 1.57 | 3.83 | 2.63 | 3.80 | 5.35 | 2.24 | 0.61 |
| **% of all volatiles** | 6.19% | 3.83% | 4.21% | 4.23% | 2.49% | 3.31% | 7.55% | 5.47% | 3.32% | 4.90% | 2.99% | 1.29% |

[1] Abbreviations are as follows: GL4—malt from green lentil germinated 4 days, GL5—malt from green lentil germinated 5 days, GL6—malt from green lentil germinated 6 days, BRL4—malt from brown lentil germinated 4 days, BRL5—malt from brown lentil germinated 5 days, BRL6—malt from brown lentil germinated 6 days, BLL4—malt from black lentil germinated 4 days, BLL5—malt from black lentil germinated 5 days, BLL6—malt from black lentil germinated 6 days. Values are expressed as means (n = 3) ± standard deviation. Various small letters (a, b, c) indicate homogenous groups according to the variable 'variety', various capital (A, B, C) letters indicate homogenous groups according to the variable 'days of germination'(Tukey test, $\alpha$ = 0.05).

hydrogen sulphide originating from the degradation of sulphur containing amino acids [35]. During the germination, the activity of proteases and amylases increases [14], resulting in the increased concentration of reducing sugars and amino acids in the germinated lentil, which can lead to the formation of benzothiazole during the drying processes. Malting decreased concentration of propanoic acid, 2-methyl-, 3-hydroxy-2,2,4-trimethylpentyl ester and pentanoic acid, 2,2,4-trimethyl-3-carboxyisopropyl, isobutyl ester in the case of black and green lentils and increased in the case of brown malts. These compounds are not thoroughly examined in the scientific literature, albeit Zhao et al. [36] have shown, that these compounds are potential indicators of aroma deterioration.

## 4. Conclusions

This study shows that malting can be a straightforward and simple way of changing volatile composition of lentil seeds. Malting resulted in the increase of total volatiles in the black lentil malts as well as brown lentil malts and in the decrease of volatiles in the green malts. The contribution of particular chemical groups of chemical compounds to the volatilome of malts was also changed by applying malting procedure. All lentil malts were characterised with greater contribution of aldehydes in the volatilome. Malting significantly reduced the concentration and contribution of terpenes in the volatilome of green lentil malts. However, precise mechanisms, which influence the changes in the composition of volatiles in the lentil seeds during the course of malting need to be investigated further in the upcoming studies, to understand the particular factors which have an impact on the creation of the volatilome of the lentil malts.

## Author Contributions

**Conceptualization:** Alan Gasiński.

**Data curation:** Alan Gasiński.

**Formal analysis:** Alan Gasiński.

**Funding acquisition:** Alan Gasiński.

**Investigation:** Alan Gasiński.

**Methodology:** Alan Gasiński.

**Project administration:** Joanna Kawa-Rygielska.

**Software:** Alan Gasiński.

**Supervision:** Joanna Kawa-Rygielska.

**Validation:** Alan Gasiński.

**Visualization:** Alan Gasiński.

**Writing – original draft:** Alan Gasiński.

**Writing – review & editing:** Alan Gasiński, Joanna Kawa-Rygielska.

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
