## [Decision Letter · Decision Letter 0]

26 Dec 2022

PONE-D-22-17168Malting - a method for modifying volatile composition of black, brown and green lentil seeds.PLOS ONE

Dear Dr. Gasiński,

Thank you for submitting your manuscript to PLOS ONE. After careful consideration, we feel that it has merit but does not fully meet PLOS ONE’s publication criteria as it currently stands. Therefore, we invite you to submit a revised version of the manuscript that addresses the points raised during the review process.

We look forward to receiving your revised manuscript.

Kind regards,

Kuldeep Tripathi

Academic Editor

PLOS ONE

https://journals.plos.org/plosone/s/fileid=ba62/PLOSOne_formatting_sample_title_authors_affiliations.pdf.2.

“A.G received award "Innowacyjny Doktorat”, no. V from Wrocław University of Environmental and Life Sciences (Poland). Funds received were used to buy standards and analytes. APC of the article is co-financed by Wroclaw University of Environmental and Life Sciences.”

4. Thank you for stating the following in the Acknowledgments/ Funding Section of your manuscript:

“This work was supported by the Wrocław University of Environmental and Life Sciences (Poland) as the Ph.D. research program "Innowacyjny Doktorat”, no. V and APC is co-financed by Wroclaw University of Environmental and Life Sciences.”

“A.G received award "Innowacyjny Doktorat”, no. V from Wrocław University of Environmental and Life Sciences (Poland). Funds received were used to buy standards and analytes. APC of the article is co-financed by Wroclaw University of Environmental and Life Sciences.”

Additional Editor Comments:

Malting can be a very promising areas in legume R &D. However, to establish the fact with evidences, some detailed analysis is required.

1.The authors must highlight the difference between malt extracts of cereals and legumes.

2. Black, green and brown seems seedcoat colour of lentils. It is required to mention the cotyledon colour as well.

3. What are the criterias of selecting these variety?

4. Some flowchart using infographics is better way to define methodology.

5. Line 32-34, "Malting is a process which is used primarily to modify grains of barley and the main reason of barley malting is increasing its enzymatic activity due to the generation and activation of various enzymes" Why in definition only barley is mentioned?

6. Introduction to be improved by adding more info. on malting in legumes.

Reviewers' comments:

Reviewer's Responses to Questions

**Comments to the Author**

1. Is the manuscript technically sound, and do the data support the conclusions?

Reviewer #1: Yes

Reviewer #2: No

2. Has the statistical analysis been performed appropriately and rigorously? 

Reviewer #1: Yes

Reviewer #2: Yes

3. Have the authors made all data underlying the findings in their manuscript fully available?

Reviewer #1: Yes

Reviewer #2: Yes

4. Is the manuscript presented in an intelligible fashion and written in standard English?

Reviewer #1: No

Reviewer #2: Yes

5. Review Comments to the Author

Reviewer #1: Dear Authors,

We all know that malting could well be utilized as a pre-treatment method to improve the nutritional properties of various grains. So, in terms of content, I found your research article very interesting. So, to guarantee its publication, you should incorporate the suggested revisions.

1. Abstract, Line 13-14, could you please explain? Try to end abstract on some conclusive note, highlighting the importance in future studies.

2. Introduction, Line 19-20, sentence seems to be irrelevant as it is not making any sense. Modify.

3. Results and discussion, A figure could be added to differentiate normal and malted lentils. A comprehensive approach explaining methodology and results in the form of a flow diagram could make this study more impactful.

5. Conclusions, Line 305-306, contradictory to abstract. Is there some typographic error or else?

6. References, re-check for journal abbreviations and follow journal style as per guidelines.

7. Grammatical errors, there is always a scope for improvement. Check whole article thoroughly and remove language errors to improve the overall quality.

Reviewer #2: This paper is having a very routine information on biochemical constituents of malted lentil. The relevance of the results or the contribution of identified constituents for ensuring superior malting quality is not explained. In this context, it is not clear how the information is practically useful. The statement in abstract section w.r.t the results is also not matching with the actual results.

6. PLOS authors have the option to publish the peer review history of their article (what does this mean?). If published, this will include your full peer review and any attached files.

Reviewer #1: No

Reviewer #2: **Yes: **Sherry Rachel Jacob

---

## [Author Response · Author response to Decision Letter 0]

7 Mar 2023

Dear Reviewers.

Our answers can be found in the main manuscript files, added as a seperate file.

Best regards

---

## [Decision Letter · Decision Letter 1]

17 Jul 2023

PONE-D-22-17168R1Malting - a method for modifying volatile composition of black, brown and green lentil seeds.PLOS ONE

Dear Dr. Gasiński,

Thank you for submitting your manuscript to PLOS ONE. After careful consideration, we feel that it has merit but does not fully meet PLOS ONE’s publication criteria as it currently stands. Therefore, we invite you to submit a revised version of the manuscript that addresses the points raised during the review process.

You are requested to address the response of reviewer 3. The author must address the queries related to methodologies, replicates and germination time. However, authors are requested to add suitable ad new references of lentil and malting. The authors should also include a para comprising economic value of malt prepared from different sources.

We look forward to receiving your revised manuscript.

Kind regards,

Kuldeep Tripathi

Academic Editor

PLOS ONE

Journal Requirements:

Additional Editor Comments:

Dear Authors, Realizing the new area of research, I recommend for minor revision. Please address the queries of reviewer 3.

Reviewers' comments:

Reviewer's Responses to Questions

**Comments to the Author**

1. If the authors have adequately addressed your comments raised in a previous round of review and you feel that this manuscript is now acceptable for publication, you may indicate that here to bypass the “Comments to the Author” section, enter your conflict of interest statement in the “Confidential to Editor” section, and submit your "Accept" recommendation.

Reviewer #3: (No Response)

Reviewer #4: (No Response)

2. Is the manuscript technically sound, and do the data support the conclusions?

Reviewer #3: Yes

Reviewer #4: Yes

3. Has the statistical analysis been performed appropriately and rigorously? 

Reviewer #3: N/A

Reviewer #4: Yes

4. Have the authors made all data underlying the findings in their manuscript fully available?

Reviewer #3: Yes

Reviewer #4: Yes

5. Is the manuscript presented in an intelligible fashion and written in standard English?

Reviewer #3: Yes

Reviewer #4: Yes

6. Review Comments to the Author

Reviewer #3: The manuscript entitled “Malting - a method for modifying volatile composition of black, brown and green lentil seeds” is well written. This study will widen the industrial use of lentils as malting improves their aroma. However, this manuscript may be accepted after the following minor revisions:

1. In the result and discussion section, the author should discuss why the total concentration of volatiles increased in green lentils but decreased in the case of black and brown lentils after the malting procedure. Some references should be added pertaining to the relation of seed colour with the concentration of volatiles.

2. In the method section, authors should add information related to the no. of replicates (biological and technical replicates) used in this study.

3. It is mentioned in the manuscript that in the malting procedure, lentil seeds are germinated for 4 days, 5 days, and 6 days. The authors should also incorporate the criteria for selecting the germination time.

Reviewer #4: I am satisfied with the scientific quality of the manuscript. The effect of malting on flavour inducing volatiles is an interesting area and this research manuscript has high scientific merit.

7. PLOS authors have the option to publish the peer review history of their article (what does this mean?). If published, this will include your full peer review and any attached files.

Reviewer #3: No

Reviewer #4: No

---

## [Author Response · Author response to Decision Letter 1]

19 Jul 2023

Dear Reviewers,

answer is provided in the separate file connected with the manuscript and figures.

Best regards

---

## [Editor Report · Decision Letter 2]

9 Aug 2023

PONE-D-22-17168R2Malting - a method for modifying volatile composition of black, brown and green lentil seeds.PLOS ONE

Dear Dr. Gasiński,

Thank you for submitting your manuscript to PLOS ONE. After careful consideration, we feel that it has merit but does not fully meet PLOS ONE’s publication criteria as it currently stands. Therefore, we invite you to submit a revised version of the manuscript that addresses the points raised during the review process.

There are some more revision need to be addressed. 1.Language needs to be improved for more clarity like in line no. 20, 25 etc. 2. In Line no. 23, it is written that lentil is warm season crop. However, lentil is cool season legume. 3. Seeds are procured from pvt company. Can you mention that whether seeds are freshly harvested or store. If sored then how long it is stored? 4. Line no. 32, Authors defined malting in context of barley. Try to avoid that barley in definition. 5. Line no. 61, please clarify and check cotyledon is red or orange? 6. Colour of fig. 1 is not professional. Keep it simple 7. Line no. 110 use eight in place of 8. Check and correct throughout MS. Digit less than 10 to be written in alphabets.This MS may be accepted as short communication after incorporating corrections suggested by referee and editor.

We look forward to receiving your revised manuscript.

Kind regards,

Kuldeep Tripathi

Academic Editor

PLOS ONE

Journal Requirements:

Additional Editor Comments:

There are some more revision need to be addressed.

1.Language needs to be improved for more clarity like in line no. 20, 25 etc.

2. In Line no. 23, it is written that lentil is warm season crop. However, lentil is cool season legume.

3. Seeds are procured from pvt company. Can you mention that whether seeds are freshly harvested or store. If sored then how long it is stored?

4. Line no. 32, Authors defined malting in context of barley. Try to avoid that barley in definition.

5. Line no. 61, please clarify and check cotyledon is red or orange?

6. Colour of fig. 1 is not professional. Keep it simple

7. Line no. 110 use eight in place of 8. Check and correct throughout MS. Digit less than 10 to be written in alphabets

This MS may be accepted as short communication after incorporating corrections suggested by referee and editor.

---

## [Author Response · Author response to Decision Letter 2]

11 Aug 2023

Dear Editor,

answers to your queries are given in the file "Answers to the Editor". Thank your for your help in improving our manuscript.

Best regards

---

## [Editor Report · Decision Letter 3]

14 Aug 2023

Malting - a method for modifying volatile composition of black, brown and green lentil seeds.

PONE-D-22-17168R3

Dear Dr. Gasiński,

We’re pleased to inform you that your manuscript has been judged scientifically suitable for publication and will be formally accepted for publication once it meets all outstanding technical requirements.

Kind regards,

Kuldeep Tripathi

Academic Editor

PLOS ONE

Additional Editor Comments (optional):

Thanks for revision.
---

## [Editor Report · Acceptance letter]

24 Aug 2023

PONE-D-22-17168R3 

Malting - a method for modifying volatile composition of black, brown and green lentil seeds. 

Dear Dr. Gasiński:

I'm pleased to inform you that your manuscript has been deemed suitable for publication in PLOS ONE. Congratulations! Your manuscript is now with our production department. 

Kind regards, 

on behalf of

Dr. Kuldeep Tripathi 

Academic Editor

PLOS ONE